# Identifying Anthropogenic Sources of Heavy Metals in Alpine Peatlands over the Past 150 Years: Examples from Typical Peatlands in Altay Mountains, Northwest China

**DOI:** 10.3390/ijerph20065013

**Published:** 2023-03-12

**Authors:** Nana Luo, Rui Yu, Bolong Wen, Xiaoyu Li, Xingtu Liu, Xiujun Li

**Affiliations:** Key Laboratory of Wetland Ecology and Environment, Northeast Institute of Geography and Agroecology, Chinese Academy of Sciences, Changchun 130102, China

**Keywords:** heavy metals, anthropogenic sources, positive matrix factorization model, Altay Mountains, alpine peatlands

## Abstract

Alpine mountain peatlands are valuable archives of climatic and anthropogenic impact. However, the impacts of human activities on the Altay peatlands are poorly documented. Therefore, studying heavy metal (HM) concentrations, evaluating HM pollution levels, and identifying the sources in the Altay Mountain peatlands are crucial for revealing the intensity of human activities. The present study was performed on two peatland profiles: Jiadengyu (JDY) and Heihu (HH). The contents of HM and ^210^Pb and ^137^Cs dating technologies were used to construct a profile of anthropogenic pollutant distributions in the peatlands. Furthermore, the enrichment factor (EF) and geo-accumulation index (Igeo) of selected HMs were used to evaluate the risk assessment of HMs. The association of metals and assignment of their probable sources were examined using principal component analysis (PCA) and a positive matrix factorization model (PMF). The results showed that the concentrations of elements Cu, Zn, Cr, Pb, Ni, and As were at high levels in the two peatlands of the Altay Mountains, while the elements Hg and Cd were in low concentrations. Moreover, the concentrations of Cu, Cd, Hg, and Sb were higher than the background values of local element and posed a high environmental risk to the ecosystem. Combined with the results of the chronology, the peatland records indicated considerable growth in HMs concentrations from 1970 to 1990 related to recent anthropogenic activities. Additionally, the main sources of HMs are mining activities, domestic waste, and traffic sources in the two peatlands. Due to the environmental protection policies implemented since 2010, the natural processes have been the primary origin of HMs in peatlands, while emissions of industrial, agricultural, and domestic waste were still fundamental sources. The results of this study describe the sedimentary features of HMs in alpine mountains, and the data provide an essential theoretical basis for the evolutionary process through the characteristics of HM deposition.

## 1. Introduction

Peatlands can record the history of environmental pollution, as they reflect human activities fairly accurately and also play a vital role in the global carbon cycle [1,2]. Since the Second Industrial Revolution, human activities have affected the balance of terrestrial ecosystems, often causing severe damage to sensitive and fragile peatlands in middle and high latitudes [3]. Due to natural disasters (erosion and fire) and strong anthropogenic disturbances (drainage, reclamation, overgrazing, and infrastructure construction), peatlands worldwide are shrinking in size and are being degraded in function, becoming a carbon source in the context of climate change [4]. Geochemical studies of historical human activities in peatlands are not confined to Europe but have been performed worldwide [5]. 

In historical and modern periods, human production and life activities inevitably generate heavy metal (HM) emissions to the atmosphere or the surrounding environment [6]. HMs, being highly toxic and difficult to degrade, can be transferred and enriched in the food chain through multiple media such as sediment and soil, thus endangering human health and causing irreversible damage to the environment [7,8]. HM sources include natural sources and man-made sources. In addition to HMs produced by nature, the strength of human activities, especially mining and metallurgical activities, is directly related to the contents of HMs in the surrounding geological carriers. Recently, studies have shown that the enrichment of Cu, Pb, and Cd caused by human activities has occurred in the Altay Mountain region and the surrounding Tuva area, where the content of HMs in the atmosphere increased due to human mining of metal deposits and manufacturing of metal implements in historical times, and the HMs eventually entered the peatlands via atmospheric deposition [9,10]. Some studies have also found that since HMs in sediments are mainly stored in the soil from the soil-forming parent material of the area in which they are located and from substances released by human activity, their contents are influenced by sediment properties [11,12]. Therefore, analyzing the records of sedimentary characteristics of HMs in peatlands and exploring the main sources can help to understand the characteristics of regional environmental changes and the extent of pollution caused by anthropogenic activities. Such information is important for taking relevant measures to prevent and control sediment pollution.

Detecting the sources of HMs is vital but complicated due to difficulties caused by various metals arising from anthropogenic and natural sources. Moreover, it is challenging to measure metal loading fluxes for all contamination sources [13]. The natural source of HMs are the weathering of rocks and forest fires. At the same time, anthropogenic sources primarily involve mining and smelting, electronics, agriculture, sewage sludge, and fossil fuel combustion. Currently, there are various receptor models to detect the primary origins of HMs, including principal component analysis (PCA), cluster analysis (CA), multiple line regression, and parameter analysis [14,15,16]. These approaches can divide HMs into various categories according to their spatial concentration patterns. In addition, HMs in similar categories can be attributed to the same sources [17,18]. Nevertheless, these approaches cannot measure the proportions of all metal categories. The U.S. Environmental Protection Agency (US EPA) has presented a useful source–receptor model called the positive matrix factorization mode (PMF) that can measure the environmental contaminants’ source contributions and distributions [18,19]. The quantification studies generally decompose the primary dataset into a contribution matrix according to factor profiles. Furthermore, the contribution source parameters of all contaminants can determine their nature and origin. Limited results indicate the superiority of PMF results to other statistical approaches such as PCA [20]. The main advantage of the PMF model is that it employs individual error estimates that allow handling of outliers and discrete data such as missing values and values below the detection limit (BDL) [21]. HMs contamination assessment approaches such as enrichment coefficient (EF) and ground accumulation index (Igeo) can detect the natural and anthropogenic origins of HMs and evaluate their pollution levels [22].

The Altay Mountains in the arid and semi-arid area of northwest China have a complex history of socio-economic growth [23]. Recent studies on the sedimentary records of HMs in peatlands have focused on the peatlands in the Great and Little Xing’an Mountains in the northeast and the Zoige region in the Qinghai–Tibet Plateau [24,25,26,27]. In contrast, fewer research works exist on the sedimentary records of peatlands in the Altay Mountains in Xinjiang focusing on long time scales (e.g., the Holocene period) [28,29,30]. As a first attempt to contribute regional information and fill current knowledge gaps, the main purposes of the study are (1) exploring the history of environmental pollution recorded in peatland sediments using an analysis of HM concentrations and (2) further analyzing the sources and contributions of contaminants in sediments by combining PCA and PMF models.

## 2. Materials and Methods

### 2.1. Regional Setting and Samples Collection

The Altay Mountains in China are located in the northernmost part of the Xinjiang Uygur Autonomous area, between 85°31′37″–91°1′15″ E and 46°30′35″–49°10′45″ N. The Altay Mountains are in a mid-continental position, and the climate is cold in a continental temperate zone. According to the annual meteorological data from the Kanas meteorological station in the Altay Mountain Area of Xinjiang, the average temperatures range from −16 °C to −12 °C in January and do not exceed 16 °C in July. The annual rainfall happens irregularly during the year, mainly from June to August, and its average is 350–600 mm from west to east. According to the elevation of the peatlands, the intensity of disturbance by human activities, and the thickness of the peatlands, the study mainly selected the representative peatland of Jiadengyu and Heihu (indicated by JDY and HH, respectively) as the main analysis objects. The JDY peatland is highly disturbed by human activities and has a low altitude, with roads, gas stations, and large parking lots around the sampling sites, and the HH peatlands have a high altitude and less human interference activities, which is a sensitive area of climate change.

The peatland of JDY is herbaceous peat, and the peatland of HH is a frost swelling peat mound, and the vegetation types are mainly Carex and Mos, which is a typical herbaceous-peat moss bog [31]. The samples were collected in mid-August 2018, and the locations of the sampling sites are shown in Figure 1 and Appendix A. Two parallel peat cores were drilled in the vicinity of the main sample site to ensure sufficient sample volume for the analysis. In addition, to ensure the accuracy of the study and improve the temporal resolution, the depth of the peatland profile at JDY is 60 cm, and the depth of the peatland profile at HH is 30 cm. the sample site was stratified with a stainless-steel bread knife to cut the collected peat column cores from top to bottom for 1 cm sampling, and 60 and 30 samples were obtained from the two peat column cores, respectively, for a total of 90 samples. All samples were stored in polyethylene bags and transmitted to the laboratory for cryopreservation. 

### 2.2. Chronology

^210^Pb and ^137^Cs data were employed to derive core chronologies and related sedimentation rates. Count periods for ^210^Pb were generally in the interval of 50,000–86,000 s, providing a measurement accuracy between ca. 65% and 610% at the 95% confidence level. The detailed dating method has been described in Reference [32].

### 2.3. Physicochemical Analysis

According to the Chinese standard methods (GB/T 17138-1997, GB/T 17141-1997), Pb, Cr, Cu, Zn, Cd, and Mn in air-dried dusts were digested by HClO_4_-HNO_3_-HF. All acids used were of ultrapure grade. For the solutions obtained by digestion, when a flame (air acetylene) atomic absorption spectrophotometer, FAAS (AA-6300C, Shimadzu, Kyoto, Japan), was insufficiently sensitive for measurement, HM concentrations were determined by a graphite furnace atomizer (EX7i, Shimadzu, Japan). Briefly, 0.10 g air-dried dust samples were digested using the method of V_2_O_5_(25 mg) + HNO_3_(5 mL) + H_2_SO_4_(2 mL) in 240 °C. The concentrations of Hg, As, and Sb in extracts were determined by atomic fluorescence spectrometer (AFS, PF6-2, PGENERAL, Beijing, China). The detection limits for Pb, Cr, Cu, Zn, Cd, and Mn were 1, 4, 2, 2, 0.001, and 2 mg kg^−1^, respectively. The standard reference material [GBW 07405 (GSS-5)] was obtained from the Center of National Standard Reference Material of China. The recovery rates were 95–105%. All the analyses were carried out in triplicate, and analytical reagent blanks were carried through the sample preparation and analytical process [33]. 

### 2.4. Contamination Assessment Methods

#### 2.4.1. Enrichment Factors

The EF has been extensively utilized to evaluate the impact of anthropogenic activities on the HM concentrations in sediment through estimating the metals’ relative contribution acquired from anthropogenic origins compared with the corresponding values from natural origins [29,34]. In order to attain the relative degree of metal contamination, comparisons with background concentrations in the Earth’s crust were performed. Due to its abundance in nature and its uniform concentration, Titanium (Ti) is generally employed for normalizing reference elements. Pollution can be categorized into five groups based on the EF value. EF < 2, 2 < EF < 5, 5 < EF < 20, 20 < EF < 40, and EF > 40 describe minimum enrichment, moderate enrichment, significant enrichment, very high enrichment, and extremely high enrichment, respectively. Buatmenard and Chesselet (1979) [35] first presented the EF calculation as follows:(1)$$\mathrm{EF}=\frac{\left(\mathrm{Metal}/\mathrm{Ti}\right)\mathrm{sample}}{\left(\mathrm{Metal}/\mathrm{Ti}\right)\mathrm{background}}$$

EF < 1 indicates that a metal principally originates from natural origins, while EF > 1 describes anthropogenic contamination: the higher the EF value, the more significant the anthropogenic activities’ contribution.

#### 2.4.2. The Geo-Accumulation Index

Muller [36] first described the geo-accumulation index (Igeo) as a measure used to assess lake sediment pollution by comparing the analyzed HMs contents with the background crustal mean elemental concentrations. The current work employed the following equation to derive Igeo for the analyzed metals:(2)$${\mathrm{Igeo}=\log}_{2}\mathrm{Cn}÷\left(1.5\times \mathrm{Bn}\right)$$
where Cn represents the element’s measured concentration in the tested sediment (surface soils), and Bn describes the element’s geochemical background value in fossil argillaceous sediment (continental crust mean or mean shale). The constant 1.5 is employed to alleviate the impact of potential changes in the background values caused by lithological variation in the sediments [37]. The mean abundance of chemical elements in the Earth’s crust [38] is adopted as the geochemical background value. According to Rao et al. [39], seven classes of Igeo were determined: uncontaminated (Igeo < 0), uncontaminated to moderately contaminated (0 < Igeo < 1), moderately contaminated (1 < Igeo < 2), moderately to heavily contaminated (2 < Igeo < 3), heavily contaminated (3 < Igeo < 4), heavily to extremely contaminated (4 < Igeo < 5), and extremely contaminated (Igeo > 5).

### 2.5. PMF Model

The PMF model is a standard model employed to evaluate the HMs’ quantitative source allocation [40,41]. It is based on decomposing the dataset into matrices via the following relation:(3)$$X_{\mathrm{ij}}=\sum_{k=1}^{p}g_{ik}f_{jk}+e_{ij}$$
where each element of the HM concentration matrix for the ith sampling position and jth HM is described by X_ij_; the source number is represented by p; g_ik_ indicates the factor contributing to the sample; f_jk_ describes the factor contributing to the pollution origin; and e_ij_ represents the residual matrix. The values of g_ik_ and f_jk_ can be derived by evaluating the model object function (Q) [42], as described by the subsequent relation:(4)$$Q={\sum_{i=1}^{n}\sum_{j=1}^{m}\left(\frac{e_{ij}}{u_{ij}}\right)}^{2}$$
where u_ij_ describes the uncertainty in the jth HMs for the ith sample. Two input files, the sample species content and uncertainty, are necessary to start the PMF model [42]. The following equation can derive the uncertainty of the species content (u):(5)$$\mathrm{Uij}=5/6\times \mathrm{MDL}\;\;\;\;(X_{\mathrm{ij}}\leq MDL)$$
(6)$$U_{ij}=\sqrt{\left(\sigma \times c\right)+{\left(0.5\times MDL\right)}^{2}}\;\;\;\;X_{\mathrm{ij}}\geq MDL$$

Sample concentration and uncertainty are two necessary datasets for the PMF model. In the current work, the concentrations of all samples exceeded the detection limit, and the uncertainty value could be obtained through Equation (5).

### 2.6. Data Analysis

The dimensionality of a dataset should be reduced to simplify its description. PCA was utilized to derive the relevant factors according to geochemical data references [17,18]. A correlation matrix analysis was employed to derive the correlations between various elements. The factor analysis was accomplished via SPSS 22. For the present study, 15 HMs of sediment samples in two peatlands were integrated into the PMF model. The current work utilized the EPA PMF model (ver. 5.0) software. The detailed data for the software handling, involving data management, and uncertainty computations have been presented in the literature (U.S. EPA, 2019). SPSS 22.0 was employed for the data analysis. Pearson’s correlation analysis was used to detect EFs related to each metal in sediment samples. The sampling site map (Figure 1) was plotted via Arc GIS 10.4, and the other figures were generated using Origin 2021.

## 3. Results

### 3.1. Sediment Chronology and Deposition Rate

A dating analysis of the peat profile at JDY has been carried out by Luo et al. [32]. The activity of ^210^Pb radioactivity of the HH peatland profiles showed a gradual decrease with depth (Figure 2). Approximately 60 cm of the JDY peatland was deposited in the last 135 years, corresponding to 1883–2018. The analysis for the HH peatland showed that 30 cm has been deposited during approximately 186 years (1832–2018). The deposition rate and deposition flux of peatlands can be calculated by combining ^210^Pb dating and dry bulk density. The sedimentation rates (SR) of JDY and HH have changed considerably over the past 100 years or so, with means of 0.15 cm yr^−1^ and 0.03 cm yr^−1^, respectively, and average sedimentation fluxes of 0.025 g cm^−2^ yr^−1^ and 0.003 g cm^−2^ yr^−1^, respectively.

### 3.2. Depth Distribution of Elements’ Concentration

The variations of 15 elements in the peatland of JDY are shown in Figure 3. The mean values of HMs contents from high to low were Al: 41,984 (mg·kg^−1^) > Fe: 16,449.3 (mg·kg^−1^) > Ca: 13,311.1 (mg·kg^−1^) > Ti: 1507.1 (mg·kg^−1^) > Mn: 237.27 (mg·kg^−1^) > V: 72.8 (mg·kg^−1^) > Zn: 46.2 (mg·kg^−1^) > Cu: 37.6 (mg·kg^−1^) > Cr: 28.5 (mg·kg^−1^) > Pb: 25.6 (mg·kg^−1^) > Ni: 19.6 (mg·kg^−1^) > As: 3.8 (mg·kg^−1^) > Sb: 0.67 (mg·kg^−1^) > Hg: 0.26 (mg·kg^−1^). The depth profile can be divided into three sections: 1–18-centimeter layer where all HMs exhibited a decreasing trend; 19–47-centimeter layer where all exhibited an increasing trend; and 47–60-centimeter layer showing different trends, i.e., elements of As, Zn, Hg, Al, Cr, Fe, Ti, Mn, and V showed decreasing trends, while Sb, Cu, Pb, Cd, and Ni exhibited increasing trends. Combining the age–depth model, the peatlands sequence can be divided into three stages. Stage I: Before 1959, elements of As, Zn, Hg, Al, Cr, Fe, Ti, Mn, and V exhibited increasing trends, while Sb, Cu, Pb, Cd, and Ni exhibited decreasing trends. Stage II: The elements in each profile showed a decreasing trend during 1960–1998, and the contents of various types of HMs had higher fluctuations before 1989, where the values were smaller and fluctuated less after 1989. Stage III: After 1998, an increasing trend was observed with more erratic fluctuations.

In HH peatlands, the average values of HMs content from high to low were Al: 22,046 (mg·kg^−1^) > Fe: 10,804 (mg·kg^−1^) > Ca: 10,774 (mg·kg^−1^) > Ti: 980.6 (mg·kg^−1^) > Mn: 106.7 (mg·kg^−1^) > Cu: 66.2 (mg·kg^−1^) > Zn: 43.5 (mg·kg^−1^) > Ni: 39.9 (mg·kg^−1^) > V: 38.7 (mg·kg^−1^) > Cr: 20 (mg·kg^−1^) > Pb: 6.2 (mg·kg^−1^) > As: 3.5 (mg·kg^−1^) > Sb: 0.96 (mg·kg^−1^) > Cd: 0.25 (mg·kg^−1^) > Hg: 0.23 (mg·kg^−1^) (Figure 3). As, Zn, Cr, Hg, Al, Fe, Ti, Mn, and V showed decreasing trends, while the rest of the elements exhibited increasing trends. Specifically, greater fluctuation was observed at the surface layer of 1–7 cm; values in the 8–25-centimeter layer fluctuated less, with lower averages, and below 25 cm there was greater fluctuation, where Cr was increasing and Cd and As exhibited more consistent trends. Although the 1–7-centimeter layer showed an overall decreasing trend with higher values, after 8–25 cm the values dropped with fewer fluctuations, and the trends of Cd and As were more consistent. Combined with the chronological analysis results, we concluded that Cu, Pb, As, Sb, Cd, Ni, and Ca exhibited increasing trends before 1904, while Zn, Cr, Hg, Mn, V, Al, Fe, and Ti showed decreasing trends. Between 1905 and 1960, Cu, Cr, Pb, Sb, As, Cd, V, Al, and Ti showed decreasing trends, while Ni, Ca, Zn, Hg, Mn, and Fe exhibited increasing trends. After 1960, each element showed an increasing trend, except for Hg and Ni, which exhibited slightly decreasing trends with more significant values.

The deposition rates of HMs profiles can be calculated by combining the peat framework (Table 1). The average accumulation rates of Cd were 0.001 mg·kg^−1^ yr^−1^ and 0.006 mg·kg^−1^ yr^−1^; the average accumulation rates of Zn were 0.164 mg·kg^−1^ yr^−1^ and 1.25 mg·kg^−1^ yr^−1^; the average accumulation rates of Cr were 0.075 mg·kg^−1^ yr^−1^ and 0.765 mg·kg^−1^ yr^−1^; and the average accumulation rates of Hg were 0.01 mg·kg^−1^ yr^−1^ and 0.06 mg·kg^−1^ yr^−1^. The accumulation rates of HMs in profile of JDY were higher than those in the peatland of HH. The distribution trends of the accumulation rate of each HMs with depth in the JDY and HH profiles are shown in Appendix A. The accumulation rates of elements and concentrations are consistent with the trends of the profiles, among which the accumulation rates of As are larger in the surface layer and smaller in the bottom layer and become larger over time. The accumulation rates of Cd and Hg tend to increase over time.

### 3.3. Risk Evaluation of HMs in Sediments

EF and Igeo values in the peatlands are shown in Figure 4. The JDY peatland showed that the EF values of Mn, As, Ni, Zn, Cr, and Ca were below 2, a minimal enrichment level caused by the Earth’s crust. The EF values of Fe, Al, V, Pb, and Cu ranged from 2 to 5, showing moderate enrichment, indicating both anthropogenic and natural sources. The EF values of Cd, Sb, and Hg ranged from 5 to 20, indicating a significant enrichment. The EF values in the HH peatland cores are presented in Figure 4. The EF values of Zn, Ca, As, Fe, V, Cr, Pb, Al, and Mn elements were less than 2 for a minimal enrichment degree, and they mainly originate from the Earth’s crust. The EF values of Ni, Cu, and Cd elements ranged from 2 to 5, showing moderate enrichment and indicating anthropogenic and natural sources. The EF values of Sb and Hg ranged from 5 to 20, indicating a significant enrichment. Combined with the Igeo index classification and pollution degree analysis, we concluded that Hg in JDY peatlands showed moderate pollution; Cd and Sb exhibited unpolluted–medium pollution degree, and the remaining elements showed no pollution status. Hg in HH peatlands showed moderate pollution; Cu, Cd, and Sb elements showed unpolluted–medium pollution levels, and the other elements exhibited no pollution status.

### 3.4. Source Apportionment of HMs in Sedimentary Cores

The PMF model and PCA method were utilized to investigate the levels of HMs in the sediments in JDY and HH peatlands and the contribution of each source to the sediment content of each HM. The input data of the PMF model were composed of HM contents and uncertainty data. The residual control was achieved by the sequential running of the model and setting the number of factors to find the minimum Q value. The results indicated that the best simulation was obtained when there were four factors and the seed number was set to 86, when the rotation factor Fpeak was set to 0.5, and when the number of iterations was 20. The residual errors were normally distributed between 3 and −3, and the fitting coefficients R2 were all above 0.8. The difference between the Q value and its theoretical value was less than 10%, indicating that the model analysis met the requirements. Among the 15 HMs, Cr was set as bad. The remaining 14 HMs were classified as a strong category (S/N > 9), and the contributions of various origins to the HMs are presented in Figure 5a,b.

The PMF results for JDY are presented in Figure 5a. The contribution of F1 reached 17.15%, a level that was compatible with the PCA 4 group result (Appendix A). Moreover, the load elements As, Cd, Pb, and Cu had high values. The peatlands of JDY are in a scenic tourist area that is disturbed to a greater extent due to frequent tourism activities, and large amounts of As and Cd pollution are generated in the long run. Therefore, F1 was classified as a “traffic source” from tourism activities; the contribution value of F2 reached 24.15%, compatible with the PCA 3 result (Appendix A), and the main elements were Hg, Sb, and Al in the order of source contribution value. Hg was closely related to tourism activities. The sampling site of JDY is in the tourist attraction area, and the roadsides are all areas with widespread tourism activities subject to a more significant degree of disturbance that can generate a large amount of Hg pollution in the long run; the pollutants include toilet paper, glass bottles, plastic products, foam, and other such solid waste produced by tourism activities. Therefore, F2 can be classified as a “domestic waste pollution source.” The contribution value of F3 was 32.24%, compatible with the PCA 1 results (Appendix A). The main elements were Zn, Mn, Fe, Cd, and Ti. The Zn, Cd, and Fe are from mining and smelting of mineral resources. Moreover, the destruction of vegetation and the transport of soil layers, which expose soil layers rich in Zn and Cd that accumulate due to heavy rain and natural transport processes, indicated that F3 originated from mining and agricultural activities. The contribution value of F4 was 26.45%, compatible with PCA 2 (Appendix A), and the main elements in the order of source contribution value were Sb, Ca, Al, Pb, V, and Ti. Therefore, F4 can be attributed to geological background sources.

Figure 5b displays the contributions of various sources to the HMs in HH peatlands. The PMF model results revealed that F1 was ranked according to the size of source contribution values as Cu, Ni, Ca, Cr, and As and was compatible with PCA 3 (Appendix A). The high contents of Cr and Cu may be due to tourism driving the tertiary industry development, generating a large amount of domestic pollution. Therefore, F1 can be categorized as a domestic waste pollution source. F2 was in order of the source contributions of Cd, Sb, Pb, and V. Cd and Pb also originate from automobile exhaust emissions, oil spills, and rubber tire wear. Zn is an additive used for automobile tire hardness and is often employed as a trace additive in livestock feed. It is also a cause of soil Zn enrichment in transportation activities and agricultural fertilization. The HH peatlands are an essential source of Cd, Cu, Pb, and Zn in the agricultural and pastoral areas and industrial emissions from mining processes. F2 can be classified as an industrial–agricultural–transportation complex source. The main elements of F3 are Hg, Zn, Mn, and Ca, compatible with the PCA 4 results (Appendix A). Hg is an atmospheric transport element. The Altay Mountain area is the most important mining area in China. As an associated element of Zn ore, Hg is exposed to the atmosphere during the mining process and is accumulated in the area by atmospheric deposition. Therefore, F3 was analyzed as an atmospheric deposition source. The main elements of F4 were Mn, V, Fe, Al, and Ti, compatible with the PCA 1 results (Appendix A). The EF results show that these elements have low enrichment and do not exceed the fundamental background values, indicating that they originated from crustal weathering. Therefore, F4 can be classified as a geological background source.

## 4. Discussion

The Xinjiang soil’s background values, the national soil environmental quality standard (GB15618-1995), and the background values of Altay area soil were utilized to determine the pollution levels of peatlands [43,44]. As presented in Appendix A, Cu, Cd, Hg, and Sb elements exceeded the Altay area’s background values. The overall values of HMs in the JDY peatlands were higher than those in the HH peatlands, and Pb in the JDY exceeded its local background values. Furthermore, the elements in excess of background levels represented different anthropogenic sources. There was a considerable positive correlation (*p* < 0.05) between the elements of Cu, Cd, Hg, and Sb (Appendix A), indicating their similar origins. Combined with the PCA and PMF models, Cu, Cd, and Pb were classified as “industrial-agricultural-transportation complex sources.” Additionally, combined with the EF and Igeo, the elements were moderately enriched or even significantly enriched, and the pollution level reached moderate and was especially serious in the JDY peatlands. This was largely due to the geographical location. The JDY peatlands are in a mountainous basin and valley area surrounded by large parking lots, gas stations, and major tourist traffic routes. Therefore, human activities (such as burning coal, motor vehicle exhaust, and industrial pollution) have had a more prominent influence on HMs.

The results of the study show that the sources of HMs are varied, primarily from natural sources (such as elements in the Earth’s crust), followed by human activities (burning coal, motor vehicle exhaust, and industrial pollution). The elements As, Zn, Hg, Cr, Al, Fe, Ti, Mn, and V showed increasing trends before 1959, and the contents of elements Cd and Hg showed increasing trends during 1960–1998. At the same time, the contents of each element during 1960–1989 were much higher than during 1990–1998. Elements Cu, Pb, and Cd showed a downward trend after 2015; during the period from 1905 to 1960, Cu, Cr, Pb, Sb, As, Cd, V, Al, and Ti showed decreasing trends, while Ni, Ca, Zn, Hg, Mn, and Fe showed increasing trends. After 1960, Hg and Ni showed decreasing trends, while the other elements showed increasing trends. Second, the historical characteristics of the deposition of HMs were further analyzed in combination with the population, cultivated land area, and coal burning amount from the statistical yearbook of the Altay region (Figure 6). We obtained from the analysis that since the founding of New China in 1949 (the reform and opening), the population and cultivated land area both showed increasing trends, and the industrial coal consumption showed an increasing trend. However, the amount of coal burning for transportation and living activities showed a downward trend after 2000, a result that was in sync with China’s economic development [45,46]. Further, based on the analysis of the deposition characteristics of HMs in peatlands, after 2010, due to the environmental protection policy, the HMs in the two peatlands of the Altay Mountains showed lower values, a result that was consistent with the present research results. This also indicated that human activities have influenced the deposition characteristics of HMs.

Studies have shown that Pb generally originates from gasoline combustion, pesticides, and fertilizer use [25,47]. Hg, Cu, and Cd are elements representative of agricultural activities [48,49]. The tourism development has led to the tertiary industry development and generated domestic garbage, thereby depositing Cu and Ni. Additionally, the high number of tourists inevitably increases HM sources (plastic cups, drinks, plastic bags, and batteries). Tourists bring their food when they visit, and their scattered food residues have high organic matter content that makes the soil acidic after decaying and will release more HMs upon contact with discarded metal products [50]. The HMs are released upon contact with discarded metal products. The research results were compatible with those of [8,51]. Wang et al. [50] found that regional random factors (agricultural irrigation, traffic, domestic waste pollution, and industrial production) influence HMs in the Altay region, and the influencing factors are complex, with both anthropogenic and natural sources. Zn, Cr, and Sb were present in relatively low amounts, while Cd and Hg had slightly higher pollution index values.

Turdi et al. [10] indicated that the enrichment of elements Cu, Pb, and Cd caused by human activities corresponds to the Altay massif and the surrounding Tuva region, where the HM elements in the atmosphere are increased due to human mining of metal deposits and manufacturing of metal implements in historical times. This is more in line with the results of this study. The Cu, Cd, Sb, and Hg trends in the JDY and HH peatlands were relatively consistent with Cu and Cd and showed decreasing trends from 1973 to 1998 and an overall increasing trend after 1998. Hg exhibited a decreasing trend from 1972 to 1989 and a higher overall value and a steady development trend after 1990. The Sb in the JDY peatlands exhibited a weak increasing trend from 1969 to 2004, a rapidly decreasing trend from 2004 to 2009, and a rapidly increasing trend after 2010. The Sb in the HH peatlands exhibited a fluctuating trend with high values before 1946 and a steady increasing trend after 1946. Combined with the general background of Xinjiang’s economic development and the Altay region’s social development, the Sb in the HH peatlands exhibited a steady increasing trend after 1946. Historical analysis showed that the higher values of HMs in the Altay region from 1950 to 1980 indicated the steady development of agriculture and animal husbandry in the Altay Mountains and its surrounding areas with the establishment of New China.

Before the reform and opening up in 1980, due to the slow socio-economic development of Xinjiang, the higher HMs values were primarily due to geological background sources, and the higher fluctuations were due to the strong anthropogenic disturbance of the peatlands’ accumulation layer caused by reclamation. Afterward, China’s economy developed rapidly, and Xinjiang’s economy was also given importance, leading to the emergence of industrial activities such as mining in the Altay region while actively developing the tertiary industry and driving the rapid development of the construction and transportation industries that promoted local economic growth [52]. This resulted in the HMs exceeding the background values and generating moderate pollution levels. Research statistics indicate that the Altay and even Xinjiang regions’ industrial, construction, and transportation industries brought in increased GDP during this period [53,54]. However, the government environmental protection policies have decreased the overall HM concentrations. Therefore, the HMs can be controlled by local environmental influences at different times with different sources, and the changes in these elements are related to the agricultural and industrial development of the surrounding areas, the implementation of emission reduction policies, and technological innovations concerning combustion.

## 5. Conclusions

Alpine peatlands sediment records, source allocation, and risk assessment of HMs in the JDY and HH peatlands in northwest China were examined to verify the effect of anthropogenic activities on the peatlands. There was little difference historically between the sedimentation rates of JDY and HH. The contents of most HMs in HH peatland were lower than those of JDY. HMs (Cd, Hg, Pb, Cu, Pb, and Sb) in the peatlands exceeded the background values. Anthropogenic loadings influenced metals (Hg, Sb, Cd, Cu, Pb, and Ni) with higher EF values. According to the Igeo analyses, sediments were significantly polluted with Hg and Sb. In addition, the two peatlands exhibited evident phase features regarding their sedimentary history. Before 1970, both peatlands were influenced by natural inputs. Between 1970 and 2010, similar HM pollution was identified, largely caused by anthropogenic activities. Since 1990, natural processes have been identified as the main origin of HMs at HH. However, the effects of anthropogenic origins, as the essential sources, cannot be neglected. The HMs in the two peatlands were influenced by anthropogenic activity. The source analysis indicated that the HMs in the peatlands of the Altay Mountains have various origins, including domestic waste, traffic sources, and industrial emissions and construction. Thus, policymakers should establish significant operations for urgent management and appropriate implementing of strategies to avoid metal contamination risks to protect the wetlands.

## Figures and Tables

**Figure 1 ijerph-20-05013-f001:**
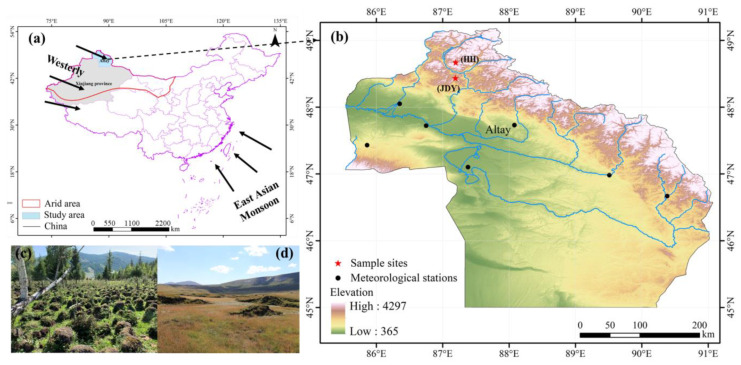
Geographical position of the sampling sites and an environmental overview of the peatlands ((**a**) denotes a map of China showing the locations of Jiadengyu and Heihu in northwest China and the Asian monsoon and the westerly; (**b**) represents the locations of Jiadengyu and Heihu peatlands (red five-pointed stars); (**c**,**d**) denote the photos showing the summer scene in the studied peatlands in August 2019).

**Figure 2 ijerph-20-05013-f002:**
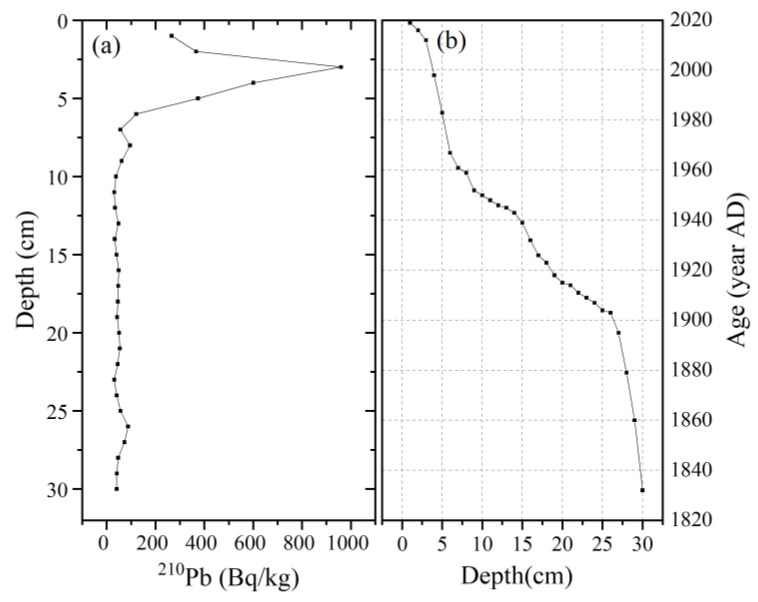
Variation of ^210^Pb (**a**) and the CRS calculated age (**b**) at the HH peatlands.

**Figure 3 ijerph-20-05013-f003:**
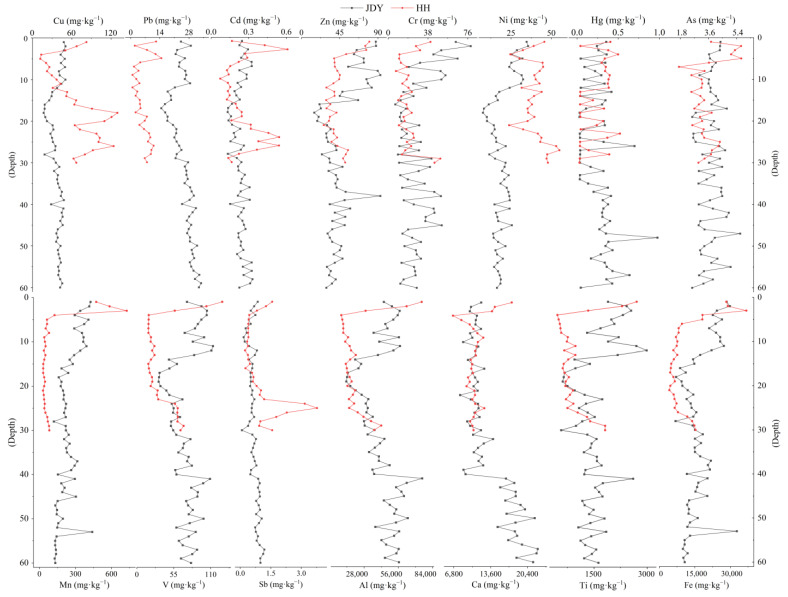
Depth variation characteristics of HMs in JDY and HH peatlands.

**Figure 4 ijerph-20-05013-f004:**
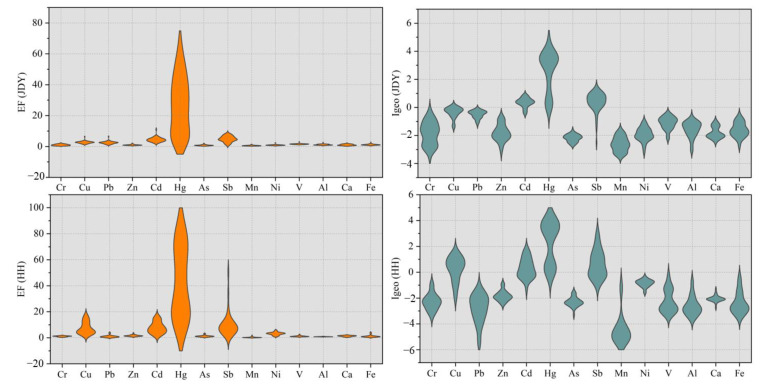
EF and Igeo index variation characteristics profiles in JDY and HH peatlands.

**Figure 5 ijerph-20-05013-f005:**
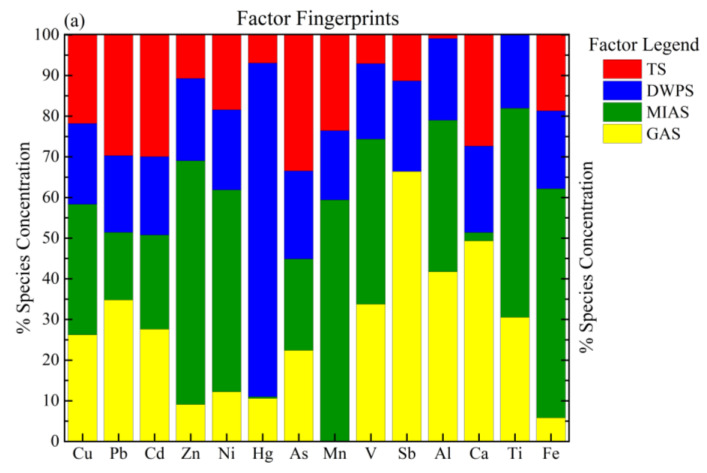
Factor profiles exhibiting the metal concentration percentages acquired from the PMF model via various origins in JDY (**a**) and HH (**b**) peatlands. TS represents traffic sources; DWPS: domestic waste pollution source; MIAS: mining industry and agricultural activities; GBS: geological background source; IATCS: industrial-agricultural-transportation complex source; ADS: atmospheric deposition source.

**Figure 6 ijerph-20-05013-f006:**
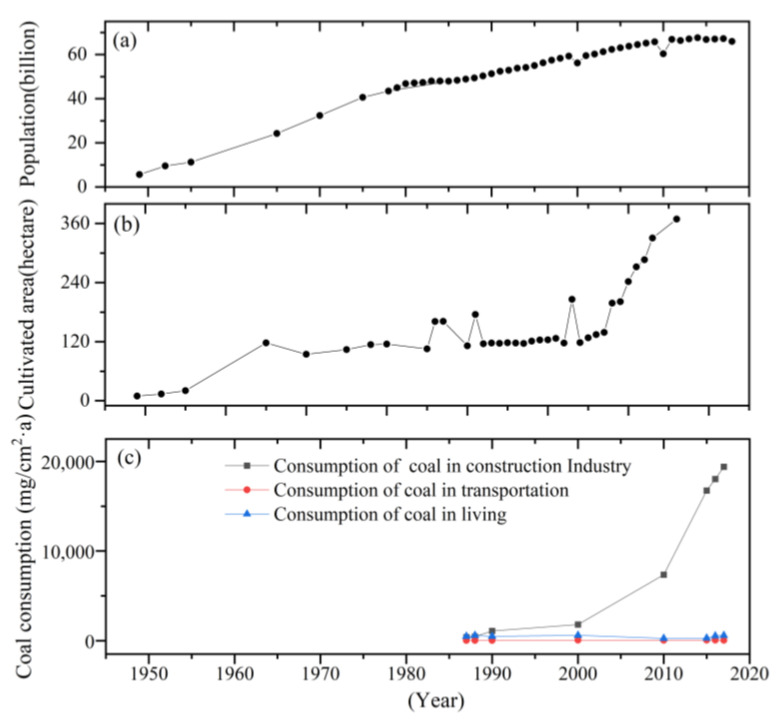
The population (**a**), cultivated area (**b**), and coal factors (**c**) (industry, transportation, and living consumption) of the Altay region.

**Table 1 ijerph-20-05013-t001:** Summary of the accumulation rate of HMs for two core samples from JDY and HH peatlands in Altay Mountains.

Elements	JDY (mg·kg^−1^ yr^−1^) (n = 60)	HH (mg·kg^−1^ yr^−1^) (n = 30)
Cu	0.951	0.229
Pb	0.635	0.021
As	0.095	0.013
Cd	0.006	0.001
Zn	1.250	0.164
Cr	0.765	0.075
Ni	0.520	0.158
Hg	0.006	0.001
Mn	6.393	0.613
V	1.855	0.167
Fe	439.371	46.679
Sb	0.016	0.003
Al	1043.284	96.317
Ca	314.176	38.755

## Data Availability

Not applicable.

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
