# Peer review of "Identifying Anthropogenic Sources of Heavy Metals in Alpine Peatlands over the Past 150 Years: Examples from Typical Peatlands in Altay Mountains, Northwest China"

_ijerph, 2023, doi:10.3390/ijerph20065013_

Round 1
Reviewer 1 Report
Dear Authors, regarding the manuscript entitled 'Identifying anthropogenic sources of heavy metals in alpine peatlands over the past 150 years: examples from typical peat-lands in Altay Mountains, northwest China' my opinion is that after some minor revisions it deserves to be accepted for publication.
More specifically, I have the following comments:
Lines 19-20 you are referring to As twice, probably this is a typographic mistake, please confirm.
Lines 97 please replace the phrase ‘purposes of the work are’ with a more formal one.
Lines 107 I think ranges rather than ‘change’ is more accurate.
Line 220 please rewrite the sentence in a more formal way.
Line 247 ‘The average values…’ I think here the soil you are referring is missing.
Lines 383-385 Please add some references to support your statement.
Line 409 Also rewrite the sentence ‘some tourists..’ in a more formal way
Line 413 please add in the text the name of the author you are referring to.
Finally homogenize the units in the text and figures, for instance mg kg-1 rather mg/kg in figure 3 and the same comment goes for the whole manuscript.
Author Response
Independent Review Report, Reviewer 1
Thank you for your support. We sincerely appreciate you for taking time to review our manuscript and provide the helpful comments and suggestions, all of which have been incorporated in the revised manuscript (The color mark of the first Reviewer Comments is purple, the color mark of the second Reviewer Comments is red, the color mark of the third Reviewer Comments is green and the color mark of the fourth Reviewer Comments is blue) as described below.
- Lines 19-20 you are referring to As twice, probably this is a typographic mistake, please confirm.
Response: We are sorry for our not exactly expression, we revised it in the revised manuscript as follows: (highlighted in Purple, Page 2, lines 19-20)
“The results showed that the concentrations of elements Cu, Zn, Cr, Pb, Ni, and As were at high levels in the two peatlands of the Altay Mountains, while the elements Hg and Cd were in low concentrations.”
- Lines 97 please replace the phrase ‘purposes of the work are’ with a more formal one.
Response: Thank you for your carefully check and revised suggestion, we have checked and revised in the revised manuscript as follows: (highlighted in Purple, Page 2, line 94)
“the main purpose of the study are (1) exploring the history of environmental pollution recorded in peatland sediments using an analysis of HM concentrations, and (2) further analyzing the sources and contributions of contaminants in sediments by combining PCA and PMF models.”
- Lines 107 I think ranges rather than ‘change’ is more accurate.
Response: Thank you for your support. We have revised the word in the revised manuscript (highlighted in Purple, Page 3, line 104)
- Line 220 please rewrite the sentence in a more formal way.
Response: Thank you for your suggestion. We have revised the sentence in the revised manuscript as follows:(highlighted in Purple, Page 5, line 211)
“A dating analysis of the peat profile at JDY has been carried out by Luo et al., [32].”
- Line 247 ‘The average values…’ I think here the soil you are referring is missing.
Response: Thanks for your valuable comment. We have revised the sentence in the revised manuscript as follows:(highlighted in Purple, Page 6, line 239)
“In HH peatlands, the average values of HMs content from high to low were…”
- Lines 383-385 Please add some references to support your statement.
Response: Thank you for pointing this out. After our examination and verification, we found that lines 383-385 were indeed the results of our study. We could invert the characteristics of element deposition according to the time when the profile was established, and thus the intensity of human activities in the historical period. However, we still revised the discussion part to make our manuscript more convincing.
- Line 409 Also rewrite the sentence ‘some tourists..’ in a more formal way.
Response: Thank you for pointing this out, we changed 'some tourists' to' tourists' in the revised manuscript. (highlighted in Purple, Page 12, line 401)
- Line 413 please add in the text the name of the author you are referring to.
Response: We are sorry for our not exactly expression, We have revised the sentence in the revised manuscript as follows:(highlighted in Purple, Page 12, lines 405-408)
“Wang et al.,[50] found that regional random factors (agricultural irrigation, traffic, domes-tic waste pollution, and industrial production) influence HMs in the Altay region, and the influencing factors are complex, with both anthropogenic and natural sources. Zn, Cr, and Sb were present in relatively low amounts, while Cd and Hg had slightly higher pollution index values.”
- Finally homogenize the units in the text and figures, for instance mg kg-1 rather mg/kg in figure 3 and the same comment goes for the whole manuscript.
Response: Thank you for your support. We are sorry for our not exactly expression and we have checked and revised in the manuscript as follows: (highlighted in Purple).
We really appreciate your patience and professional suggestions.
Best wishes.

Reviewer 2 Report
Reviewer
MDPI – International Journal of Environmental Research and Public Health
Manuscript Number: ijerph-2244921
Title: «Identifying Anthropogenic Sources of Heavy Metals in Alpine Peatlands Over the Past 150 Years: Examples from Typical Peatlands in Altay Mountains, Northwest China».
It is necessary to indicate in the methodology why exactly the isotopes of lead-210 and cesium-137 were used to estimate the age of peatlands.
line 145 Add a reference to formula 1 in the text.
line 174 Add reference to formula 2 in the text.
line 180 Add a reference to formula 3 in the text.
line 194 Add a link to formula 4 in the text.
line 199 Add a reference to formula 5 in the text.
line 202 Add reference to formula 6, 7 in the text.
I recommend the article for publication after revision.
Author Response
Independent Review Report, Reviewer 2
Thank you for your support. We sincerely appreciate you for taking time to review our manuscript and provide the helpful comments and suggestions, all of which have been incorporated in the revised manuscript (The color mark of the first Reviewer Comments is purple, the color mark of the second Reviewer Comments is red, the color mark of the third Reviewer Comments is green and the color mark of the fourth Reviewer Comments is blue) as described below.
It is necessary to indicate in the methodology why exactly the isotopes of lead-210 and cesium-137 were used to estimate the age of peatlands.
Response: Thanks for your valuable comment. The 210Pb and 137Cs is one of the most conventional methods used in peatlands to judge the sedimentary age of a profile. Since all analyses are based on the construction of chronological profiles.
- line 145 Add a reference to formula 1 in the text.
Response: We sincerely thank you for your carefully check and revised suggestion, we have revised the Chronology in the revised manuscript as follows: (highlighted in red, Pages 2-3, lines 64-95)
“210Pb and 137Cs data were employed to derive core chronologies and related sedimen-tation rates. Count periods for 210Pb were generally in the interval of 50,000 - 86,000 s, providing a measurement accuracy between ca. 65% and 610% at the 95% confidence level. The detailed dating method has been described in Reference [32].”
line 174 Add reference to formula 2 in the text.
Response: We have done.
line 180 Add a reference to formula 3 in the text.
Response: We have revised.
line 194 Add a link to formula 4 in the text.
Response: We sincerely thank you for your carefully check and revised suggestion, we have done.
line 199 Add a reference to formula 5 in the text.
Response: We have done.
line 202 Add reference to formula 6, 7 in the text.
Response: We have revised.
We really appreciate your patience and professional suggestions.
Best wishes.

Reviewer 3 Report
This study focused on "Identifying Anthropogenic Sources of Heavy Metals in Alpine Peatlands Over the Past 150 Years: Examples from Typical Peatlands in Altay Mountains, Northwest China" The paper could be interesting for other researchers. However, some corrections are needed before the final decision.
- The abstract is too long. It should be re-organized to reflect the paper contents and main results.
- At present, the introduction wrote in generally. It needs more highlights for the novelty and the weakness in the previous studies. More important, what the reasons make you must to do the research in this area. And what are the specific contributions can be got from your research. The introduction should be re-organized.
- Please describe the drilling process, especially, using tools, method, layer structure, and so on.
- Please describe the initial assumptions made of the modeling (PCA and PMF).
- Please compare the results about deposition rate to the previous researches in the area with the similar geology background.
- ‘3.2 Depth Distribution of Elements’ Concentration’ is very difficult to read, please re-organized to reflect the main results and reasons. Moreover, please compare the results with previous researches.
- The author claimed to use the PCA and PMF models to do the source apportionment of heavy metals. Why the author needs to use two models? The results from the two models are the same? What the connection of the results from the two models?
- The sampling sites of the soil core is not too far from each other, which showed in Fig.1, but the source of heavy metals is very difficult. Why?
- Why the author claimed ‘The peatlands of JDY are in a scenic tourist area that is disturbed to a greater extent due to frequent tourism activities, and large amounts of As and Cd pollution are generated in the long run. Therefore, F1 was classified as a “traffic source” from tourism activities’? Did you do investigations about the local traffic or other research could support your results? Because, the As and Cd pollution can also from other sources.
- The data for the source apportionment of heavy metals is from the surface soil core, or the average results from the whole soil core or from which layer of the soil core. Because in different times the source of heavy metals maybe different.
- Based on the questions above, the ‘Discussion’ and ‘Conclusions’ should be re-organized too.
- Refs should be updated.
Author Response
Independent Review Report, Reviewer 3
Thank you for your support. We sincerely appreciate you for taking time to review our manuscript and provide the helpful comments and suggestions, all of which have been incorporated in the revised manuscript (The color mark of the first Reviewer Comments is purple, the color mark of the second Reviewer Comments is red, the color mark of the third Reviewer Comments is green and the color mark of the fourth Reviewer Comments is blue) as described below.
-The abstract is too long. It should be re-organized to reflect the paper contents and main results.
Response: We sincerely thank you for your carefully check and revised suggestion, we have added and revised necessary research significance in the abstract in the revised manuscript as follows: (highlighted in green, Page 1, lines 9-29)
“Alpine mountain peatlands are valuable archives of climatic and anthropogenic impact. However, the impacts of human activities on the Altay peatlands are poorly documented. Therefore, studying HMs (HMs) concentrations, evaluating HMs pollution levels, and identifying the sources in the Altay Mountain peatlands are crucial for revealing the intensity of human activities. The present study was performed on two peatland profiles: Jiadengyu (JDY) and Heihu (HH). The contents of HMs and 210Pb and 137Cs dating technologies were used to construct a profile of anthropogenic pollutant distributions in the peatlands. Furthermore, the enrichment factor (EF) and geo-accumulation index (Igeo) of selected HMs were used to evaluate the risk assessment of HMs. The association of metals and assignment of their probable sources were examined using principal component analysis (PCA) and a positive matrix factorization model (PMF). The results showed that the concentrations of elements Cu, Zn, Cr, Pb, Ni, and As were at high levels in the two peatlands of the Altay Mountains, while the elements Hg, As, and Cd were in low concentrations. Moreover, the concentrations of Cu, Cd, Hg, and Sb were higher than the background values of local element and posed a high environmental risk to the ecosystem. Combined with the results of the chronological, the peatland records indicated considerable growth in HMs concentrations from 1970 to 1990 related to recent anthropogenic activities. Additionally, the main sources of HMs are mining activities, domestic waste, and traffic sources in the two peatlands. Due to the environmental protection policies implemented since 2010, the natural processes have been the primary origin of HMs in peatlands, while emissions of industrial, agricultural, and domestic waste were still fundamental sources. The results of this study describe the sedimentary features of HMs in alpine mountains and the data provide an essential theoretical basis for the evolutionary process through the characteristics of HM deposition.”
- At present, the introduction wrote in generally. It needs more highlights for the novelty and the weakness in the previous studies. More important, what the reasons make you must to do the research in this area. And what are the specific contributions can be got from your research. The introduction should be re-organized.
Response: Thanks for your valuable comment. Both in historical times and in modern times, human production and life are bound to emit HMs into the atmosphere or the surrounding environment. Since HMs are highly toxic and difficult to degrade, they can be transferred and enriched in the food chain through multiple media such as sediments and soils, thus endangering human health and causing irreversible harm to the environment. Since the HMs in sediments mainly come from the soil-forming parent material of the area they are located and the release of human production and life stored in the soil, their contents are easily influenced by the nature of sediments. Therefore, analyzing the characteristics of sediment record information of heavy metal elements in sediment and exploring their main source status can help to understand the characteristics of regional environmental changes and the degree of pollution caused by anthropogenic activities and is important for taking relevant measures to prevent and control sediment pollution. Sediment plays the role of the ultimate receiver of pollutants in the wetland ecosystem and is an important part of the wetland system. In specific environments sediments may also release pollutants to water bodies and become a secondary pollution endogenous source. Sources of HMs include both natural and anthropogenic sources. Therefore, it is particularly important to accurately discern the sources of HMs in sediments and to quantify them to determine the relative contribution of pollutants. The typical mountain peatlands in the Altay Mountains of northwestern China, which are influenced by climate and human activities in a single way, are very sensitive to climate and environmental changes. The study reveals the impact of human activities on the peatlands ecosystem and enriches the study of peatland sedimentation processes. we have revised in the revised manuscript: (highlighted in green, Pages 1-2, lines 34-97)
- Please describe the drilling process, especially, using tools, method, layer structure, and so on.
Response: Thank you very much for your advice. As for mudstone sample collection and sample site description, we have added Table 1 based on Figure 1, hoping to grasp the general situation of sample sites as much as possible. We have added and revised in the revised manuscript as follows: (highlighted in green, Page 3, lines 116-124)
“The samples were collected in mid-August 2018, and the locations of the sampling sites are shown in Figure 1 and Table S1. Two parallel peat cores were drilled in the vicinity of the main sample site to ensure sufficient sample volume for the analysis. In addition, to ensure the accuracy of the study and improve the temporal resolution, the depth of the peatland profile at JDY is 60cm, and the depth of the peatland profile at HH is 30cm. the sample site was stratified with a stainless-steel bread knife to cut the collected peat column cores from top to bottom for 1 cm sampling, and 60 and 30 samples were obtained from the two peat column cores, respectively, for a total of 90 sam-ples. All samples were stored in polyethylene bags and transmitted to the laboratory for cryopreservation.”
Table S1. Overview of typical Peatlands profiles in the Altay Mountains
|
Sampling site |
Altitude (m) |
Peat profile characteristics |
|
JDY |
1402 |
The thickness of the profile is 60 cm, from top to bottom it can be divided as follows: turf layer: 0 - 7 cm, vegetation cover is about 80%. Grass roots are dense, containing green plant bodies; humus layer: 7 - 17 cm, poor decomposition, mainly incompletely decomposed mosses and mosses, dense modern plant roots, dark brown loamy soil with more roots; peat layer: dark brown, containing more capillary roots; 17 - 21 cm, black brown powdery clay with less roots; 21 - 30 cm brown clay, basically no roots below 30 cm. |
|
(HH) |
2190 |
The thickness of the profile is 30 cm, from top to bottom it can be divided into: turf layer: 0 - 5 cm, vegetation cover is about 80%. Grass roots are dense, containing green plant bodies; humus layer: 5 - 15 cm, poorly decomposed, mainly incompletely decomposed mosses and mosses, dense modern plant roots, dark brown loamy soil with more roots; peat layer: dark brown, loamy clay to clay, blocky, containing more capillary roots; 15 - 30 cm is brown peat layer. |
-Please describe the initial assumptions made of the modeling (PCA and PMF).
Response: Thank you for pointing this out. Since PCA is one of the commonly used methods for statistical analysis of data, it is not described in this manuscript, and we have added the key arithmetic parts about the PMF model in the manuscript. PCA:The basic idea of the PCA algorithm is to try to replace the original indicators with a new set of mutually unrelated composite indicators by regrouping a large number of indicators with certain relevance (e.g., P indicators). The usual mathematical treatment is to make a linear combination of the original P indicators as a new composite indicator. Typically, this is expressed as the variance of F1 (the first linear combination selected, i.e., the first composite indicator), i.e., the larger Var(F1) indicates that F1 contains more information. Therefore, the F1 selected among all linear combinations should have the largest variance, so F1 is called the first principal component. If the first principal component is not enough to represent the information of the original P indicators, then consider selecting F2, that is, selecting the second linear combination, in order to effectively reflect the original information, the existing information of F1 does not need to appear in F2 again, expressed in mathematical language is to require Cov (F1, F2) = 0, then call F2 as the second principal component, and so on can construct the third, fourth ........... , and the Pth principal component. It should be noted that principal component analysis is often not an end in itself, but a means to an end, and therefore it is mostly used in some intermediate part of a large research project. If it is used in multiple regression, it gives rise to principal component regression, which has excellent properties and, in addition, is very useful in compression, feature extraction and classification applications.
PCA solution: roots of characteristic equations
In linear algebra, the PCA problem can be described in the following form:
Find a matrix P consisting of a set of orthogonal bases with Y=PX such that CY 1n-1 YYT is a diagonal array. Then the row vector of P (which is a set of orthogonal bases) is the principal element vector of the data X.
Derivation of CY:
CY=YYT
=(PX)(PX)T
=PXXTPT
=P(XXT)PT
CY=PAPT
Define A=XXT, then A is a symmetric array. Diagonalizing A to obtain the eigenvectors gives:
A=EDET
Then D is a diagonal array, while E is a matrix lined up with the eigenvectors of the symmetric array A.
The point to be made here is that A is an m × m matrix, and it will have r(rm) eigenvectors. Where r is the rank of the matrix A . If r < m, then A is a degenerate array. At this time, the decomposed eigenvectors cannot cover the whole m-space. At this point, it is only necessary to obtain any m-r dimensional orthogonal vectors to fill the spaces of R in the remaining space, while ensuring the orthogonality of the basis. They will have no effect on the results. This is because the eigenvalues corresponding to these eigenvectors, i.e., the variance values, are zero.
After finding the eigenvector matrix we take P-ET, then A=PTDP, and from the linear algebra we know that the P matrix has the property P-1=PT, thus performing the following calculation:
CY=PAPT
=P(PTDP)PT
=(PPT)D(PPT)
=(PP-1)D(PP-1)
CY=D
We can see that P is the transformed base we need to find at this point. At this point we can obtain the result of PCA:
The principal element of X is the eigenvector of XXT, which is the row vector of the matrix P.
The ith element on the diagonal of the matrix CY is the variance of the data X in the direction Pi.
We can obtain the general steps of PCA solution:
1) Collect data to form a matrix of m×n. m is the number of observed variables, n is the number of sampling points.
2) Subtract the mean value of the observed variable from each observed variable (matrix row vector) to obtain the matrix X.
3) To decompose XXT, the eigenvectors and the corresponding eigenroots are obtained.
PMF model:
Receptor models are mathematical approaches for quantifying the contribution of sources to samples based on the composition or fingerprints of the sources. The composition or speciation is determined using analytical methods appropriate for the media, and key species or combinations of species are needed to separate impacts. A speciated data set can be viewed as a data matrix X of i by j dimensions, in which i number of samples and j chemical species were measured, with uncertainties u. The goal of receptor models is to solve the chemical mass balance (CMB) between measured species concentrations and source profiles, as shown in Equation 1-1, with number of factors p, the species profile f of each source, and the amount of mass g contributed by each factor to each individual sample (see Equation 1-1):
where eij is the residual for each sample/species. The CMB equation can be solved using multiple models including EPA CMB, EPA Unmix, and EPA Positive Matrix Factorization (PMF).
PMF is a multivariate factor analysis tool that decomposes a matrix of speciated sample data into two matrices: factor contributions (G) and factor profiles (F). These factor profiles need to be interpreted by the user to identify the source types that may be contributing to the sample using measured source profile information, and emissions or discharge inventories. The method is reviewed briefly here and described in greater detail elsewhere (Paatero and Tapper, 1994; Paatero, 1997).
Results are obtained using the constraint that no sample can have significantly negative source contributions. PMF uses both sample concentration and user-provided uncertainty associated with the sample data to weight individual points. This feature allows analysts to account for the confidence in the measurement. For example, data below detection can be retained for use in the model, with the associated uncertainty adjusted so these data points have less influence on the solution than measurements above the detection limit.
Factor contributions and profiles are derived by the PMF model minimizing the objective function Q (Equation 1-2):
Q is a critical parameter for PMF and two versions of Q are displayed for the model runs.
Q(true) is the goodness-of-fit parameter calculated including all points.
Q(robust) is the goodness-of-fit parameter calculated excluding points not fit by the model, defined as samples for which the uncertainty-scaled residual is greater than 4.
The difference between Q(true) and Q(robust) is a measure of the impact of data points with high scaled residuals. These data points may be associated with peak impacts from sources that are not consistently present during the sampling period. In addition, the uncertainties may be too high, which result in similar Q(true) and Q(robust) values because the residuals are scaled by the uncertainty.
The Uncertainty (u) of the species content is obtained using the following equation.
Uij=5/6×MDL ()
()
Sample concentration and uncertainty are two necessary data sets for PMF model.
- Please compare the results about deposition rate to the previous researches in the area with the similar geology background.
Response: During 2022, we analyzed HMs in the peatlands of Greater and Lesser Khingan Mountains in Northeast China, and obtained that the average deposition rate of peat profiles in Greater Khingan Mountains and Lesser Khingan Mountains was 0.036cm·yr-1 and 0.033cm·yr-1, respectively. The average deposition fluxes were 0.024g cm-2 yr-1 and 0.02g cm-2 yr-1, respectively. Combined with the modern and modern carbon accumulation rates reflected by different mountain peatlands profiles, it is found that the surface value of the peatlands in the northwest Altay Mountains is higher than the deep value, while the opposite is true in the Greater and Lesser Khingan Mountains in the northeast. Combined with the analysis of the chronological profile, it is concluded that the peat deposition rate in the Altay Mountains is slower than that in the Greater and Lesser Hinggan Mountains, which may be due to the different responses to climate change caused by different geographical locations. From 1980 to 2018, the fluctuation of sedimentary characteristics of peatlands in the northwest Altay Mountains may be related to abrupt climatic events or sedimentary events. At the same time, modern human activities and grazing of cattle and sheep were more serious. Historical records show that during this period, tourism and construction were developed on a large scale in Xinjiang and even the Altay Mountains, and these human activities may lead to the weakening of organic matter accumulation and the slowing down of peat development. The low content of organic matter in Northeast China may be due to the social and economic development of the region in its heyday, the development of the Great famine in the north, migration of a large number of people and other social factors. The calculated RERCA value of Bao et al(2015) found that the peat in Changbai Mountain is about 199.6 g C m-2 yr-1, which is consistent with the study in this paper on JDY peatland in Northwest China and Huihe peatland in Northeast China. Su Q et al., (2017) analyzed that the mean RERCA of Ruoergai peat was 86.12 g C m-2 yr-1. This is consistent with the mean values of the peatlands in Heihu and Yichun, which are sensitive to climate change, and the peatlands in JDY in the northwest Altay Mountains and Huihe in the northeast Greater Khingan Mountains, which are highly disturbed by human activities. The regional climatic and environmental information recorded in the HH of the northwest Altay Mountain and the peatland of Yichun in the northeast Lesser Khingan Mountains, which are sensitive to climate change, provided a more favorable condition for peat accumulation and development. However, as the peatland profile deposition rate is not the focus of this study, considering the length of the paper, it will not be described in the text, please understand. Thank you very much for your valuable comments. Since deposition rates were not the primary objective of this study, the discussion of deposition rates was not detailed in the manuscript.
- ‘3.2 Depth Distribution of Elements’ Concentration’ is very difficult to read, please re-organized to reflect the main results and reasons. Moreover, please compare the results with previous researches.
Response: Thank you for your comments. The general idea of the concentration analysis is that we first described the overall characteristics of the elemental concentration mean, followed by the analysis of the depositional characteristics of the elemental profiles, and finally, the depositional characteristics of the elements in different periods in combination with the chronology of the profiles. And we have checked and revised the 3.2 section and discussion in the revised manuscript (highlighted in green, Pages 6-7, lines 223-254; Pages 11-13, lines 355-424)
- The author claimed to use the PCA and PMF models to do the source apportionment of HMs. Why the author needs to use two models? The results from the two models are the same? What the connection of the results from the two models?
Response: With regard to the source analysis of pollutants, many studies are based on a research method, such as PCA analysis or PMF model. It is difficult to accurately grasp the source of pollutants by considering only one research method, and the accuracy of the model is also uncertain. Therefore, we choose to use multiple analysis methods to verify each other. First, we use PCA method to set the number of factors for multiple times to screen the source types of heavy metal elements, and draw the conclusion that there are four groups of types that can best explain the source. Then we use PMF model to repeatedly verify and analyze, and screen out the best source types, and use correlation analysis to further assist the results. In general, the method of this study is mutual validation to improve the accuracy of heavy metal source tracking in this study. For more specific PCA and correlation analysis, see the supplementary information.
- The sampling sites of the soil core is not too far from each other, which showed in Fig.1, but the source of HMs is very difficult. Why?
Response: Your good advice was very much appreciated. First, JDY Peatland has a low latitude, but because it is located in the necessary area of tourist attractions, surrounded by tourist vehicles, gas stations and other facilities, it has a high concentration of HMs, and the source of HMs is mainly caused by human activities, while HH Peatland has a high latitude and is far away from tourist attractions, so the traffic source accounts for a small proportion. However, since the peatland is located in the area of tourism, agriculture and animal husbandry, The accumulation of industrial emissions from processing in the mining industry is also an important source of the HMs Cd, Cu, Pb and Zn. The sources of HMs in the HH Peatland are complex, including both human activities and natural sources.
Sampling site of peatland at JDY in Altay mountain in China
- Why the author claimed ‘The peatlands of JDY are in a scenic tourist area that is disturbed to a greater extent due to frequent tourism activities, and large amounts of As and Cd pollution are generated in the long run. Therefore, F1 was classified as a “traffic source” from tourism activities’? Did you do investigations about the local traffic or other research could support your results? Because, the As and Cd pollution can also from other sources.
Response: Thank you for your valuable opinion, which is highly appreciated by us. In fact, we have also done a basic background investigation, and the transportation industry and industrial activities around JDY area have been analyzed in our discussion (Figure 6).
- The data for the source apportionment of HMs is from the surface soil core, or the average results from the whole soil core or from which layer of the soil core. Because in different times the source of HMs maybe different.
Response: Thank you very much for your opinion and we appreciate it. The whole section depth of JDY Peatland reaches 60cm, and the section depth of Heihu Peatland reaches 30cm. Then, we cut the sample according to 1cm for analysis, so as to avoid ignoring the element deposition characteristics in the core of the whole mud-carbon column. Later, we will also study the sedimentary characteristics of deeper sections, because some existing studies have found that the section depth of the Altay Peatland reaches 2-3m. At the same time, the chronological section construction is combined with 14C to trace the sedimentary characteristics of heavy metal elements in the Quaternary period, so as to better retrieve the influence of human activities or climate change in the past historical period.
- Based on the questions above, the ‘Discussion’ and ‘Conclusions’ should be re-organized too.
Response: We have done.
- Refs should be updated.
Response: Thanks for your carefully suggestion. We updated the references according to the revised draft.
We really appreciate your patience and professional suggestions.
Best wishes.

Reviewer 4 Report
1. The literature review in the introduction section is not comprehensive and does not present the current progress and scientific problems of international research on anthropogenic and natural sources of heavy metals recorded in peat, as well as the research work in the Altay region (domestic and Russian scholars), thus preventing the scientific value of the authors' research work from being reflected.
2. Analytical procedure: As the critical technique to tracking HMs pollution history from peat profile, 210Pb and 137Cs are widely applied. However, No 137Cs and 210Pb analytical procedure were introduced in section 2. And the cited article [28] is not related to the peat chronology from the study site at all. This makes dating result in Figure 2 questionable. Furthermore, AAS and FAAS procedure are not applicable for analyses of Hg, As, and Sb.
3. Enrichment factors and Geo-Accumulation Index were used to assess HMs pollution in peat profiles in the article. However, the elemental background concentration is very crucial for the assessment and significantly affects the assessment result. The sediment concentration in Xinjiang was applied in the calculation. However, Altay Mountain is also a very important mineralization region in China. As a result, the geochemical background for some HMs is likely greater than the value as revealed by concentration variation for Hg, As, Sb, and Cd.
4. Element mobility after deposition is not discussed in the article.
5. In section 3.4 Source Apportionment of HMs in Sedimentary Cores, the load elements As, Cd, Pb, Cu, and Hg are attributed to traffic source or tourism activities. This argument lacks evidence.
6. Inappropriate literature citations, e.g., citations to literature [1], [3], [22], [28], [43] are completely irrelevant to the content of the presentation.
Author Response
Independent Review Report, Reviewer 4
Thank you for your support. We sincerely appreciate you for taking time to review our manuscript and provide the helpful comments and suggestions, all of which have been incorporated in the revised manuscript (The color mark of the first Reviewer Comments is purple, the color mark of the second Reviewer Comments is red, the color mark of the third Reviewer Comments is green and the color mark of the fourth Reviewer Comments is blue) as described below.
We sincerely thank you for taking time to conduct this review process and evaluate our manuscript. Thank you so much for your comment here.
- The literature review in the introduction section is not comprehensive and does not present the current progress and scientific problems of international research on anthropogenic and natural sources of heavy metals recorded in peat, as well as the research work in the Altay region (domestic and Russian scholars), thus preventing the scientific value of the authors' research work from being reflected.
Response: We sincerely thank you for taking time to conduct this review process and evaluate our manuscript. Thank you for your carefully check and revised suggestion, we have checked and revised in the revised manuscript as follows: (highlighted in blue, Page 2, lines 51-58)
“Recently, Studies have shown that the enrichment of Cu, Pb, and Cd caused by human activities has occurred in the Altay Mountain region and the surrounding Tuva area, where the content of HMs in the atmosphere increased due to human mining of metal deposits and manufacturing of metal implements in historical times and the HMs even-tually entered the peatlands via atmospheric deposition [9-10]. Some studies have also found that since HMs in sediments are mainly stored in the soil from the soil-forming parent material of the area in which they are located and from substances released by human activity, their contents are influenced by sediment properties[11-12]”
- Analytical procedure: As the critical technique to tracking HMs pollution history from peat profile, 210Pb and 137Cs are widely applied. However, No 137Cs and 210Pb analytical procedure were introduced in section 2. And the cited article [28] is not related to the peat chronology from the study site at all. This makes dating result in Figure 2 questionable. Furthermore, AAS and FAAS procedure are not applicable for analyses of Hg, As, and Sb.
Response: We are sorry for our not exactly expression, and thank the reviewers for your careful review of the article. Since the results of the JDY peatland profile have been published in Luo et al. (2022), we directly cited the results of the reference to avoid data duplication and only listed the results of the profile age of the HH peatland in detail. We have revised all the references in the revised manuscript. About the test methods for elements Hg, As, and Sb, Briefly, 0.10 g air-dried soils were digested using the method of V2O5(25mg)+ HNO3(5ml)+H2SO4(2ml) in 240℃. The concentrations of Hg, As, and Sb in extracts were determined by atomic fluorescence spectrometer (AFS, PF6-2, PGENERAL, Beijing, China). And we have checked and revised in the revised manuscript. (highlighted in blue, Page 5, line 207; Page 5, line 207)
“According to the Chinese standard methods (GB/T 17138-1997, GB/T 17141-1997), Pb, Cr, Cu, Zn, Cd and Mn in air-dried dusts were digested by HClO4-HNO3-HF. All acids used were of ultrapure grade. For the solutions obtained by digestion, when a flame (air acetylene) atomic absorption spectrophotometer, FAAS (AA-6300C, Shimadzu, Japan), was insufficiently sensitive for measurement, HM concentrations were determined by a graphite furnace atomizer (EX7i, Shimadzu, Japan). Briefly, 0.10 g air-dried dusts were digested using the method of V2O5(25mg)+ HNO3(5ml)+H2SO4(2ml) in 240℃. The concen-trations of Hg, As, and Sb in extracts were determined by atomic fluorescence spectrome-ter (AFS, PF6-2, PGENERAL, Beijing, China). The detection limits for Pb, Cr, Cu, Zn, Cd and Mn were 1, 4, 2, 2, 0.001 and 2 mg kg-1, respectively. The standard reference material [GBW 07405 (GSS-5)] was obtained from the Center of National Standard Reference Mate-rial of China. The recovery rates were 95-105%. All the analyses were carried out in tripli-cate, and analytical reagent blanks were carried through the sample preparation and an-alytical process [33].”
“Luo N, Wen B, Bao K, Yu R, Sun J, Li X and Liu X (2022). Centennial records of Polycyclic aromatic hydrocarbons and black carbon in Altay Mountains peatlands, Xinjiang, China. Front. Ecol. Evol. 10:1046076. doi: 10.3389/fevo.2022.1046076.”
- Enrichment factors and Geo-Accumulation Index were used to assess HMs pollution in peat profiles in the article. However, the elemental background concentration is very crucial for the assessment and significantly affects the assessment result. The sediment concentration in Xinjiang was applied in the calculation. However, Altay Mountain is also a very important mineralization region in China. As a result, the geochemical background for some HMs is likely greater than the value as revealed by concentration variation for Hg, As, Sb, and Cd.
Response: Thank you very much for your valuable comment. In fact, we have conducted a literature search on the mining area in the Altay Mountains and found that our sampling sites are far away from the mining area and separated by several high mountains (the red boxes in the figure represent the mining area), while our study also showed that the concentrations of some HMs are higher than the background values of the Altay area crust (see table).
Table S4 Elemental mean and background values (units: mg kg-1)
|
Elements |
Mean value in JDY |
Mean value in HH |
Background value of soil elements |
|
Ti |
1507.18 |
980.59 |
3117 |
|
Cu |
37.60 |
66.17 |
31.02 |
|
Pb |
25.57 |
6.21 |
23.63 |
|
As |
3.75 |
3.53 |
11.2 |
|
Cd |
0.23 |
0.26 |
0.12 |
|
Zn |
46.17 |
43.51 |
99.2 |
|
Cr |
28.49 |
20.01 |
58.63 |
|
Ni |
19.62 |
39.92 |
47.16 |
|
Hg |
0.26 |
0.23 |
0.02 |
|
Mn |
237.27 |
106.73 |
907.6 |
|
V |
72.86 |
38.74 |
100 |
|
Fe |
16449.31 |
10804.14 |
30890 |
|
Sb |
0.67 |
0.96 |
0.31 |
|
Al |
41984.00 |
22046.03 |
77440 |
|
Ca |
13311.13 |
10774.29 |
29450 |
- Element mobility after deposition is not discussed in the article.
Response: Thank you for pointing this out, the main purpose of this study is to trace the information of human activities in the past historical period by using the centennial sedimentation characteristics of peatlands, and only analyzed the elemental sedimentation characteristics of peat profiles, but not the migration process after elemental sedimentation. We will also conduct an in-depth study on the transport of HMs after profile deposition by combining more peatland sample sites and deeper peat cores at a later stage.
- In section 3.4 Source Apportionment of HMs in Sedimentary Cores, the load elements As, Cd, Pb, Cu, and Hg are attributed to traffic source or tourism activities. This argument lacks evidence.
Response: Thanks for your valuable comment. Regarding the elemental source analysis we mainly used multiple analysis methods to verify each other, we firstly used PCA analysis to categorize elemental components, secondly we used PMF model to classify elemental categories, at the same time we combined with correlation analysis to reconfirm the categories between elements, and finally we judged which elemental sources have consistency. Finally, based on the results of the above studies combined with the results of previous studies and local elemental background values to determine the conclusions. The main references for the study of heavy metals in the Altay Mountains of Xinjiang are as follows:
Zhang Y, Yang P, Tong C, et al. Palynological record of Holocene vegetation and climate changes in a high-resolution peat profile from the Xinjiang Altay Mountains, northwestern China[J]. Quaternary Science Reviews, 2018, 201: 111-123.
Wang G, Late Holocene human activities recorded by peat deposits in Altai, northern Xinjiang [D] Lanzhou University, 2017.
Zhang Y, Holocene peat development characteristics and regional environmental evolution in Altay Mountains, Xinjiang [D] Graduate School of the Chinese Academy of Sciences (Northeast Institute of Geography and Agroecology), 2016.
Zhang, Z., A. Jilili, and F. Jiang, Environment risk and chemical forms of heavy metals in farmland of Ebinur Basin. Scientia Geographica Sinica, 2015. 35(009): p. 1198-1206.
Lu, X., et al., Contamination assessment of mercury and arsenic in roadway dust from Baoji, China. Atmospheric Environment, 2009. 43(15): p. 2489-2496.
Chen S., Wang M., and Li S, The status and problems of heavy metal pollution control in farmland soil in China. Earth Science Frontiers, 2019, 26(6):7.
Bao K, Shen J, Wang G, et al. Atmospheric deposition history of trace metals and metalloids for the last 200 years recorded by three peat cores in Great Hinggan Mountain, Northeast China. Atmosphere, 2015, 6(3): 380-409.
Liu Y J. Spatial distribution characteristics and pollution assessment of heavy metals Pb, Cd and Hg in soil of Huainan Mining area based on GIS [D]. Hefei University of Technology, 2016.
Sutherland, R.A., Bed sediment-associated trace metals in an urban stream, Oahu, Hawaii. Environmental Geology, 2000. 39(6): p. 611-627.
Wang W., Bai Z., Liu D., et al., Soil heavy metal pollution and its potential ecological risk assessment in Kanas Scenic area. Bulletin of Soil and Water Conservation, 2018, 38(6):352-357.
Taylor, S.R., Abundance of chemical elements in the continental crust -a- new table. Geochimica Et Cosmochimica Acta, 1964. 28(AUG): p. 1273-1285.
Zuzolo, D., et al., Assessment of potentially harmful elements pollution in the Calore River basin (Southern Italy). Environmental Geochemistry & Health, 2016. 39(3): p. 1-18.
Pu, J., et al., The spatial analysis of soil elements and a risk assessment of heavy metals based on regular methods in the Xinjiang local region. Journal of Agro-Environment Science, 2018. 37(6): p. 11.
Zhang Z., Jilili Abudu.l, Assessment of Heavy Metal Pollution in the Soil of Tianshan Mountains and Analysis of Potential Ecological Risk[J]. advances in earth science, 2014, 29(5):608-616.
A Yi Nur. Research on the development of Tourism economy in Kanas, Xinjiang [D]. Xinjiang Agricultural University, 2013.
- Inappropriate literature citations, e.g., citations to literature [1], [3], [22], [28], [43] are completely irrelevant to the content of the presentation.
Response: We are sorry for our not exactly expression, and thank the reviewers for your careful review of the article. We revised all references in the revised manuscript.
We really appreciate your patience and professional suggestions.
Best wishes.

Round 2
Reviewer 3 Report
Accept in present form
Author Response
Independent Review Report, Reviewer 3(Round 2)
Thank you for your support. We sincerely appreciate you for taking time to review our manuscript and provide the helpful comments and suggestions, all of which have been incorporated in the revised manuscript (The color mark of the Reviewer Comments is red) as described below.
(x)English language and style are fine/minor spell check required
Response: We sincerely thank you for your carefully check and revised all spell in the revised manuscript.
窗体顶端
--Are all the cited references relevant to the research?
Response: Thanks for your carefully suggestion. We updated the references according to the revised draft and delated the redundant references.
- Are the results clearly presented?
Response: Thanks for your valuable comment. We have corrected the result, mainly the language and word spelling mistakes.
--Are the conclusions supported by the results?
Response: Thank you very much for your advice. The conclusion is summarized and condensed by combining the result section.
We really appreciate your patience and professional suggestions.
Best wishes.
References
- Sun, J.-J., et al., Study of Jinchuan Mire in NE China II: Peatland development, carbon accumulation and climate change during the past 1000 years. Quaternary International, 2019. 528: 18-29.
- Bandara, S., Records of atmospheric mercury deposition and post-depositional mobility in peat permafrost archives from central and northern Yukon, Canada. 2017.
- Magiera, T., et al., Peat bogs as archives of local ore mining and smelting activities over the centuries: A case study of Miasteczko lskie (Upper Silesia, Poland). Catena, 2021. 198(21): 105063.
- Renberg, I., M.W. Persson, and O., Emteryd, Pre-industrial atmospheric lead contamination detected in Swedish lake sediments. Nature, 1994. 368(6469): 323-326.
- Hu, X., C. Wang, and L. Zou., Characteristics of heavy metals and Pb isotopic signatures in sediment cores collected from typical urban shallow lakes in Nanjing, China. Journal of Environmental Management, 2011. 92(3): 742-748.
- Song, Z., et al., Influence of flocculation conditioning on environmental risk of heavy metals in dredged sediment. Journal of Environmental Management, 2021. 297.
- Wang, M., et al., Heavy metal contamination in surface sediments from lakes and their surrounding topsoils of China. Environmental Science and Pollution Research, 2021. 28(23): 29118-29130.
- Zuzolo, D., et al., Assessment of potentially harmful elements pollution in the Calore River basin (Southern Italy). Environmental Geochemistry and Health, 2017. 39(3): 531-548.
- Du, P. and S. Tian., Mean element background of lithic debris stream sediments in Xinjiang. Geophysical and geochemical exploration, 2001. 25(002): 117-122.
- Turdi, M., Distribution characteristics of soil heavy metal content in northern slope of Tianshan Mountains and its source explanation. Chinese Journal of Eco-Agriculture, 2020. 21(7).
- Turgun, A., et al., Distribution characteristics and sources of heavy metals in soil profile of lakeside zone on the west bank of Boston lake, Xinjiang. Ecological Science, 2019. 038(006): p. 53-59.
- Liu, Z., et al., Pollution characteristics and pollution degree assessment of heavy metals is serfacc sedimeris from Boster Lake. Journal of Shihezi University (Nature Science), 2019. 37(5): 8.
- Zhang, Z.X., et al., Assessment of heavy metal contamination, distribution and source identification in the sediments from the Zijiang River, China. Science of the Total Environment, 2018. 645: 235-243.
- Li, Y.B., et al., Evaluation of the Possible Sources and Controlling Factors of Toxic Metals/Metalloids in the Florida Everglades and Their Potential Risk of Exposure. Environmental Science & Technology, 2015. 49(16): 9714-9723.
- Loska, K. and D. Wiechula., Application of principal component analysis for the estimation of source of heavy metal contamination in surface sediments from the Rybnik Reservoir. Chemosphere, 2003. 51(8): 723-733.
- Chandrasekaran, A., et al., Multivariate statistical analysis of heavy metal concentration in soils of Yelagiri Hills, Tamilnadu, India - Spectroscopical approach. Spectrochimica Acta Part a-Molecular and Biomolecular Spectroscopy, 2015. 137: 589-600.
- Xie, Z., et al., Contamination of Trace Elements in River Ecosystem and Source Apportionment Based on Their Relationship with Landscape Patterns. Polish Journal of Environmental Studies, 2021. 30(4): 3327-3339.
- S. Environmental Protection Agency., EPA Positive Matrix Factorization (PMF) 5.0, Fundamentals and User Guide, 2014.
- Tang, W.Z., et al., Heavy metal sources and associated risk in response to agricultural intensification in the estuarine sediments of Chaohu Lake Valley, East China. Journal of Hazardous Materials, 2010. 176(1-3): 945-951.
- Comero, S., et al., Characterisation of Alpine lake sediments using multivariate statistical techniques. Chemometrics and Intelligent Laboratory Systems, 2011. 107(1): 24-30.
- Magesh, N.S., et al., Hazardous heavy metals in the pristine lacustrine systems of Antarctica: Insights from PMF model and ERA techniques. Journal of Hazardous Materials, 2021. 412.
- Bing, H.J., et al., Spatial variation of heavy metal contamination in the riparian sediments after two-year flow regulation in the Three Gorges Reservoir, China. Science of the Total Environment, 2019. 649: 1004-1016.
- Wu S, Haiying, Batan., A study of ecological and environmental problems in the two river source areas of the Altay Mountains, 2003.
- Bao, K., et al., Estimates of recent Hg pollution in Northeast China using peat profiles from Great Hinggan Mountains. Environmental Earth Sciences, 2016. 75(6).
- Wang, G.P., J.S. Liu, and H, Tang., The long-term nutrient accumulation with respect to anthropogenic impacts in the sediments from two freshwater marshes (Xianghai Wetlands, Northeast China). Water Research, 2004. 38(20): 4462-4474.
- Bao, K.S., et al., Recent Carbon Accumulation in Changbai Mountain Peatlands, Northeast China. Mountain Research and Development, 2010. 30(1): 33-41.
- Yang Z., Su Q., Chen H., et al., Anthropogenic impacts recorded by a 200-year peat profile from the Zoige Peatland, northeastern Qinghai-Tibetan Plateau, Catena, 2021, 206(105463).
- Zhang Y., Yang P., Tong C., et al., Palynological record of Holocene vegetation and climate changes in a high-resolution peat profile from the Xinjiang Altay Mountains, northwestern China. Quaternary Science Reviews, 2018, 201: 111-123.
- Wang G., Late Holocene human activities recorded by peat deposits in Altay, northern Xinjiang. Lanzhou University, 2017.
- Luo, N., Yu, R., Mao, D., Wen, B., Liu, X. 2021. Spatiotemporal variations of wetlands in the northern Xinjiang with relationship to climate change. Wetlands Ecology and Management, 29: 617-631.
- Zhang Y., Holocene peat development characteristics and regional environmental evolution in Altay Mountains, Xinjiang. Graduate School of the Chinese Academy of Sciences (Northeast Institute of Geography and Agroecology), 2016.
- Luo N., Wen B., Bao K., Yu R., and Liu X., Centennial records of Polycyclic aromatic hydrocarbons and black carbon in Altay Mountains peatlands, Xinjiang, China. Front. Ecol. Evol. 2022,10:1046076.
- Cao, W., et al., Post relocation of industrial sites for decades: Ascertain sources and human risk assessment of soil polycyclic aromatic hydrocarbons. Ecotoxicology and Environmental Safety, 2020. 198: 9.
- Glasby, G.P., and P. Szefer., Marine pollution in Gdansk Bay, Puck Bay and the Vistula Lagoon, Poland: An overview. Science of the Total Environment, 1998. 212(1): 49-57.
- Buatmenard P., Chesselet R., Variable influence of the atmosphere flux on the trace-metal chemistry of oceanic suspended mater. Earth and Planetary Science Letters 1979; 42: 399-411.
- Muller, G., Heavy-metals in sediment of the rhine-changes since 1971. Umschau in Wissenschaft Und Technik, 1979. 79(24): 778-783.
- Perumal, K., J. Antony, and S. Muthuramalingam, Heavy metal pollutants and their spatial distribution in surface sediments from Thondi coast, Palk Bay, South India. Environmental Sciences Europe, 2021. 33(1): 20.
- Irabien, M.J., et al., A 130 year record of pollution in the Suances estuary (southern Bay of Biscay): Implications for environmental management. Marine Pollution Bulletin, 2008. 56(10): 1719-1727.
- Rao, K., et al., Spatial-temporal dynamics, ecological risk assessment, source identification and interactions with internal nutrients release of heavy metals in surface sediments from a large Chinese shallow lake. Chemosphere, 2021. 282: 9.
- Cheng W., Lei SG., Bian ZF et al., Geographic distribution of heavy metals and identification of their sources in soils near large, open-pit coal mines using positive matrix factorization. Journal of Hazardous Materials 2020; 387: 11.
- Corella, J.P., et al., Recent and historical pollution legacy in high altitude Lake Marboré (Central Pyrenees): A record of mining and smelting since pre-Roman times in the Iberian Peninsula. Science of The Total Environment, 2020. 751: 141557.
- Paatero P., Least squares formulation of robust non-negative factor analysis. Chemometrics and Intelligent Laboratory Systems 1997; 37: 23-35.
- China, S.S., E.P.A.o., Environmental Quality Standard for Soils. 1995.
- Zhang, Z., A. Jilili, and F. Jiang, Environment risk and chemical forms of heavy metals in farmland of Ebinur Basin. Scientia Geographica Sinica, 2015. 35(009): 1198-1206.
- Lu, X., et al., Contamination assessment of mercury and arsenic in roadway dust from Baoji, China. Atmospheric Environment, 2009. 43(15): 2489-2496.
- Chen S., Wang M., and Li S, The status and problems of heavy metal pollution control in farmland soil in China. Earth Science Frontiers, 2019, 26(6):7.
- Bao K., Shen J., Wang G, et al., Atmospheric deposition history of trace metals and metalloids for the last 200 years recorded by three peat cores in Great Hinggan Mountain, Northeast China. Atmosphere, 2015, 6(3): 380-409.
- Liu Y J., Spatial distribution characteristics and pollution assessment of heavy metals Pb, Cd and Hg in soil of Huainan Mining area based on GIS. Hefei University of Technology, 2016.
- , R.A., Bed sediment-associated trace metals in an urban stream, Oahu, Hawaii. Environmental Geology, 2000. 39(6): 611-627.
- Wang W., Bai, Liu D., et al., Soil heavy metal pollution and its potential ecological risk assessment in Kanas Scenic area. Bulletin of Soil and Water Conservation, 2018, 38(6):352-357.
- Taylor, S.R., Abundance of chemical elements in the continental crust -a- new table. Geochimica Et Cosmochimica Acta, 1964. 28(AUG): 1273-1285.
- Zuzolo, D., et al., Assessment of potentially harmful elements pollution in the Calore River basin (Southern Italy). Environmental Geochemistry & Health, 2016. 39(3): 1-18.
- Pu, J., et al., The spatial analysis of soil elements and a risk assessment of heavy metals based on regular methods in the Xinjiang local region. Journal of Agro-Environment Science, 2018. 37(6): 11.
- Zhang Z., Jilili Abudu.l, Assessment of Heavy Metal Pollution in the Soil of Tianshan Mountains and Analysis of Potential Ecological Risk. Advances in Earth Science, 2014, 29(5):608-616.
- Yi. Nur., Research on the development of Tourism economy in Kanas, Xinjiang. Xinjiang Agricultural University, 2013.

Reviewer 4 Report
The authors cited articles should be carefully checked to make sure that they are relevant to the research.
Author Response
Independent Review Report, Reviewer 4(Round 2)
Thank you for your support. We sincerely appreciate you for taking time to review our manuscript and provide the helpful comments and suggestions, all of which have been incorporated in the revised manuscript (The color mark of the Reviewer Comments is red) as described below.
(x) English language and style are fine/minor spell check required
Response: We sincerely thank you for your carefully check and revised suggestion, we have carefully revised all spell in the revised manuscript.
--Are all the cited references relevant to the research?
Response: Thanks for your carefully suggestion. We updated the references according to the revised draft and delated the redundant references.
We really appreciate your patience and professional suggestions.
Best wishes.
